# Contribution of Recycled Moisture to Precipitation in Northeastern Tibetan Plateau: A Case Study Based on Bayesian Estimation

**Xue Qiu** [1,2], **Mingjun Zhang** [2,3,*], **Zhiwen Dong** [4], **Shengjie Wang** [2,3], **Xiuxiu Yu** [2,3], **Hongfei Meng** [2,3] and **Cunwei Che** [2,3]

1   College of Geography and Environmental Engineering, Lanzhou City University, Lanzhou 730070, China; qx888@nwnu.edu.cn
2   College of Geography and Environmental Science, Northwest Normal University, Lanzhou 730070, China; wangshengjie@nwnu.edu.cn (S.W.); 2016211723@nwnu.edu.cn (X.Y.); 2016211720@nwnu.edu.cn (H.M.); 2017212151@nwnu.edu.cn (C.C.)
3   Key Laboratory of Resource Environment and Sustainable Development of Oasis, Lanzhou 730070, China
4   Cold and Arid Regions Environmental and Engineering Research Institute, Chinese Academy of Sciences, Lanzhou 730070, China; dongzhiwen@lzb.ac.cn
*   Correspondence: mjzhang@nwnu.edu.cn; Tel.: +86-0931-797-1750

**Abstract:** (1) Background: The degree to which local precipitation is supplied by recycled moisture is a reflection of land surface–atmosphere interactions and a potentially significant climate feedback mechanism. This study tries to figure out the water cycle and precipitation mechanism at a mountainous region and then provides a reference for similar mountainous regions outside China. (2) Methods: The dual-isotopes and Bayes-based program MixSIAR is used to assess contributions of advected, transpirated and evaporated vapor to local precipitation. (3) Results: The average percent contribution of recycled moisture (i.e., the sum of surface evaporated vapor and transpirated vapor) to local precipitation at the Qilian Mountains during 2017 plant growing season is about 37% (the upper quartile and the lower quartile was 30% and 43%, respectively). (4) Conclusions: Although the contribution of advection vapor dominated during the plant growing season, the contribution of recycled moisture is also important in such an alpine region. Furthermore, the commonly used simple linear mixing models often yield contributions greater than 100% or less than 0% and are likely to underestimate the contribution of recycled moisture to local precipitation. Although the alternative Bayesian model is not perfect, either, it is still a big improvement.

**Keywords:** recycled moisture; Bayesian mixing model; MixSIAR; growing season; Qilian Mountains

## 1. Introduction

Precipitation over a terrestrial region is usually considered to be derived from three main sources, i.e., water vapor advected into the region by air mass motion and water vapor supplied by surface evaporation and vegetation transpiration from the land surface of the region. The recycled moisture or recycled precipitation is defined as water vapor that is produced by surface evaporation and vegetation transpiration within a region and falls again as precipitation within the same region [1–3]. Hence, the relative contribution of recycled moisture to total precipitation, also called the recycling ratio, provides a potentially diagnostic measurement that describes land surface processes and land surface–atmosphere interactions and is of great significance in the study of hydrological cycle. The recycling ratio depends on geographic location, size of the domain under consideration, topography, climate, seasonal variation, vegetation and many other factors [1,3–6]. The global annual mean recycling ratio for 500 km scales is 9.6%, and for 1,000 km, is less than 20% [3]. Recycled moisture by evapotranspiration is probably the most important mechanism sustaining rainfall for continental catchments, particularly in semiarid areas [7].

Earlier research showed that precipitation is largely derived from external origins [8,9]. However, later studies indicated that land regions can also be significant sources of water vapor [10–12]. Portis et al. [13] found that local evaporation is a major source of moisture for small rainfall events, whereas larger events draw from both local evaporation and horizontal atmospheric water vapor flux convergence. In addition, some researchers suggested that natural or anthropogenic changes that enhance (or inhibit) convection could alter the contribution of local moisture to precipitation, such as irrigation [14,15], urban heat islands [16], planting bands of vegetation [17], large-scale deforestation [18] and land use changes [19].

The study of recycled moisture has been approached in a variety of ways, such as analytical models [1,3,19–23], numerical tracer experiments [24–33] and physical analysis using isotope data. Analysis of stable isotope composition ($\delta^{18}O$ and $\delta^2H$) of precipitation can directly be used to infer the fraction of recycled moisture in precipitation. This isotopic evaluation is usually based on the assumptions that the stable isotope composition of the water vapor that results in precipitation is a mixture of water vapors sourced from advection, evaporation and transpiration, and each source has its unique isotopic composition [34,35].

Generally speaking, the isotope method is performed using linear mixing model, i.e., the two-end-member linear mixing model [6,36,37] or the three-end-member linear mixing model [5,38,39]. If the precipitation vapor is assumed to be formed by advected vapor, evaporated vapor and transpired water, such a model is called three-end-member linear mixing model. For special surfaces, such as large water bodies or deserts without vegetation, it is assumed that transpired vapor is negligible, and such a model is called two-end-member linear mixing model. It is currently solved by direct linear equations or in the software IsoError [40], but usually, the percent contributions by this method may be greater than 100% or less than 0%; such results need further processing or parameter adjustment. However, Bayesian mixing models describe the phenomena better than simpler linear mixing models by explicitly taking into account uncertainty in source values [41,42], categorical and continuous covariates [43–45] and prior information [41]. The results based on a Bayesian mixing model are more credible, since all feasible solutions can be obtained. Since Moore and Semmens [41] first proposed the use of a Bayesian mixing model to calculate the ratio of each source in the mixture, several software have been developed, such as the MixSIR, SIAR and MixSIAR. Because both linear and Bayesian models are used to calculate the contributions of different sources to a mixture, we have used this software to research the contribution of different water vapor types to precipitation. Because MixSIAR incorporates several years of advances in Bayesian mixing model theory since MixSIR and SIAR and represents a collaborative coding project between the investigators behind MixSIR and SIAR [46], this study will use MixSIAR to calculate the contribution of different water vapor types to local precipitation.

The contributions of the different moisture sources to precipitation were calculated in the Qilian Mountains, which are the main mountains of the northeastern Tibetan Plateau. In recent decades, significant climatic and environmental changes have been observed in the Qilian Mountains, including temperature increase, glacier shrinkage and vegetation degradation [47,48]. It is important to investigate the contribution of vegetation transpiration to mountainous water cycle, and the information of local moisture recycling is helpful to understand the potential influence of water transfer projects in arid areas. In spite of previous studies in other areas in China (Table 1), the mountainous areas such as the Qilian Mountains still need a detailed assessment based on more in situ observations. In such a monsoon marginal region, the diversity of the upper wind makes this study more meaningful. Although several related studies have been conducted in this region [38,49,50], the uniqueness of this research lies in the use of Bayesian mixing model to estimate percent contributions of different water vapor to precipitation for the first time.

**Table 1.** Percent contributions of recycled moisture to local precipitation in China in previous studies.

| Study Area | Percent Contributions | | Reference |
|---|---|---|---|
| | Recycled Moisture | Advected Vapor | |
| Taiwan Island | 37% | 63% | Peng et al. (2011) [5] |
| Urumqi river basin | less than 2.0 ± 0.6% | / | Kong et al. (2013) [6] |
| Shiyang river basin | 9% for evaporation and 14% for transpiration | 77% | Li et al. (2016) [38] |
| Tianshan Mountains | 16.2% at large oases of Urumqi; less than 5% at small oases such as Caijiahu and Shihezi | / | Wang et al. (2016) [39] |
| Northwestern China | 10–14% | / | Hua et al. (2017) [46] |
| Shiyang river basin | 17% in the mountain regions, 28% in the oasis regions, 15% in the desert regions | 83% in the mountain regions, 72% in the oasis regions, 85% in the desert regions | Zhu et al. (2019) [50] |

Since the transpiration mainly occurred in the plant growing season, only this period was selected. Particularly, the dual-isotope, Bayesian mixing model were used to determine the percent contribution of advection, evaporation and transpiration vapor to local precipitation. The objective of the work described here is: (1) obtain quantitative estimates of the degree to which land–atmosphere moisture recycling is active over the Qilian Mountains and figure out the water cycle and precipitation mechanism in this region; (2) calculate the contribution of each water vapor type to precipitation using the Bayesian mixing model MixSIAR instead of linear mixing model, which avoids the unreasonable results caused by simple formula operation in the past; and (3) provide a reference about precipitation supplement mechanisms for similar mountainous regions outside China.

## 2. Materials and Methods

### 2.1. Study Area

The Qilian Mountains are located in the northeastern edge of the Tibetan Plateau (93°30′ E–103° E, 36°30′ N–39°30′ N) and consist of several northwest–southeast ranges and valleys (Figure 1). It is more than 850 km long from east to west and 200–300 km wide from north to south, with an elevation range from 3500 m to 5000 m. Under a continental climate, the annual average temperature was 3.4 °C, and the average annual precipitation amount was 329 mm during 2016–2017 (Table 2). The rivers are jointly supplemented by glacial melt water and mountain precipitation, which is considered the important water source for the oases along the Hexi Corridor on the northern slope. The vegetation and soil types show obvious zonal and vertical gradients due to the complex natural conditions. The soil types from top to bottom include alpine cold desert soil, alpine meadow soil, alpine steppe soil and mountain meadow soil. The main vegetation belts are desert steppe belt, grassland belt, forest steppe belt, shrub steppe belt, meadow steppe belt, ice and snow belt, etc. This research established six observation stations on the north and south slopes of the Qilian Mountains based on the National Meteorological Station; see Table 2 for specific information.

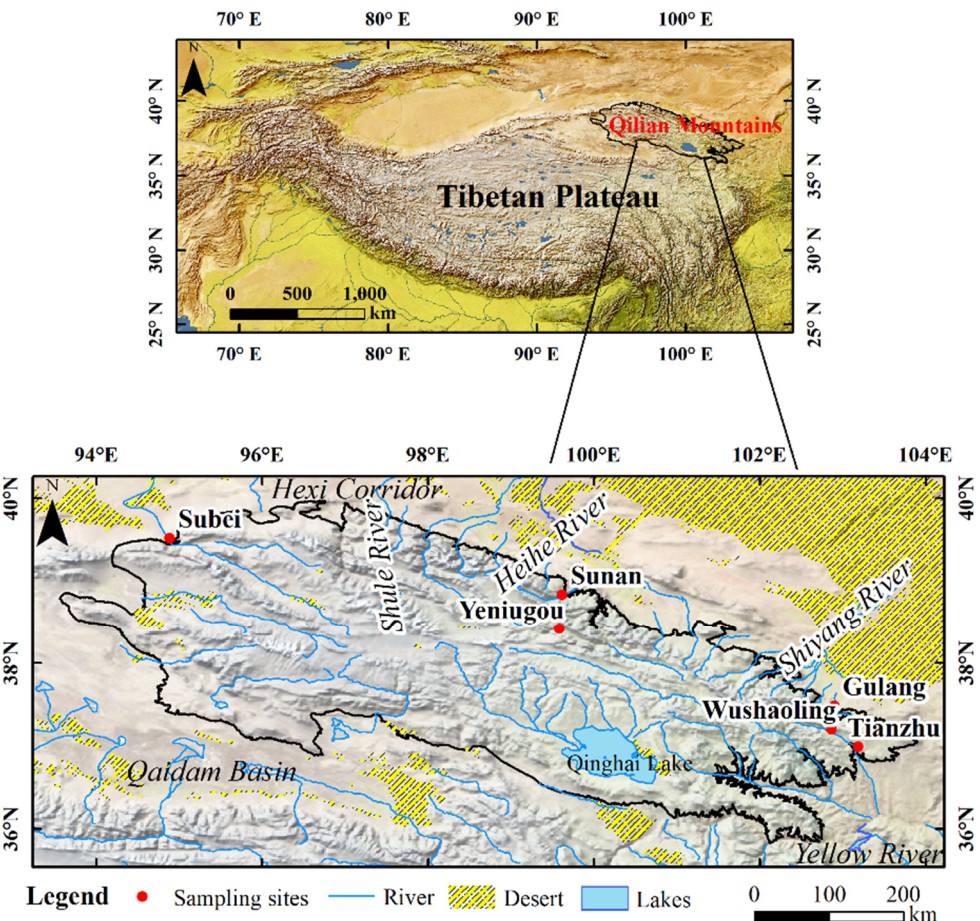

**Figure 1.** Location of sampling sites in the Qilian Mountains (black outline), northeastern Tibetan Plateau. The small map shows the Qilian Mountains (red outline) in the Tibetan Plateau.

**Table 2.** Information and meteorological parameters of sampling sites during 2016–2017 (T-Air Temperature, P-Precipitation Amount and h-Relative Humidity).

| Station | Latitude | Longitude | Elevation (m) | $T$ (°C) | $P$ (mm) | $h$ (%) |
|---|---|---|---|---|---|---|
| Subei | 39°31′ | 94°52′ | 2137 | 7.3 | 124 | 41 |
| Sunan | 38°50′ | 99°37′ | 2311 | 5.1 | 216 | 49 |
| Gulang | 37°29′ | 102°54′ | 2072 | 7.3 | 232 | 46 |
| Yeniugou | 38°25′ | 99°35′ | 3320 | −1.7 | 512 | 60 |
| Tianzhu | 36°59′ | 103°11′ | 2484 | 4.1 | 236 | 58 |
| Wushaoling | 37°12′ | 102°52′ | 3045 | 1.3 | 494 | 60 |

*2.2. Sample Collection*

The event-based precipitation samples (together with the relevant meteorological parameters, i.e., air temperature, relative humidity, precipitation amount, etc.) were collected by full-time members of meteorological stations. A clean large-mouth container was placed on flat ground before the rain started to fall, and plastic bags and HDPE bottles were prepared. Solid precipitation was put in a plastic bag until it melted at room temperature, and then it was put into bottle; liquid precipitation was directly put into bottle, and the bottle was sealed. At the same time, we recorded the relevant meteorological parameters, such as temperature, relative humidity and pressure at the beginning and end of precipitation.

The soil samples were collected monthly during the plant growing season (from May to September) in 2017. Soil samples were collected from soil pits about 1 m × 1 m × 1 m

with 10-cm intervals. The final isotope value for each sample was an arithmetic average of several repeat samples.

The pan evaporation samples were collected continuously for one month during July–August 2017. We placed an evaporation pan at each site and filled it with water. We sampled water once a day using an HDPE bottle for one month continuously. The pan experiment in July and August can better represent the average condition of the whole plant growing season.

All samples had repeats in order to calculate their average values. A total of 750 samples (excluding repeat samples) were collected. The inventory of various samples is shown in Table 3.

**Table 3.** Inventory of sample numbers collected at the Qilian Mountains.

| Sites | Precipitation | Soil | Pan Experiment |
|---|---|---|---|
| Subei | 42 | 55 | 30 |
| Sunan | 44 | 55 | 31 |
| Gulang | 19 | 55 | 29 |
| Yeniugou | 60 | 55 | 30 |
| Tianzhu | 35 | 55 | 20 |
| Wusaholing | 50 | 55 | 30 |

### 2.3. Meteorological Data

The monthly data of the pressure (surface), meridional wind, zonal wind and specific humidity (from 1000 hPa to 300 hPa level) during May–September in 2017 were acquired from ERA Interim data base of the European Centre for Medium-Range Weather Forecasts (ECMWF) (https://www.ecmwf.int/) (accessed on 11 March 2019).

### 2.4. Laboratory Analysis

First, water was drawn from soil using the Automatic Water Extraction System LI-2100 (LICA United Technology Limited, Beijing, China). More information of the analysis was detailed in Qiu et al. [51]. Then the isotope compositions ($^{18}O$ and $^2H$) in liquid water including precipitation and soil water were analyzed by the Liquid Water Isotope Analyzer DLT-100 (Los Gatos Research, California, USA) in the Stable Isotope Laboratory, College of Geography and Environmental Science, Northwest Normal University. Each sample and standard sample was injected and analyzed sequentially six times using a microliter syringe. The results of the first two analyses were discarded in order to eliminate any previous water residual and instrument memory effect. The final results of each sample were the arithmetic mean of the measurements of the last four injections. The isotope ratios of $^{18}O/^{16}O$ and $^2H/^1H$ in the water samples are expressed as $\delta^{18}O$ and $\delta^2H$, which are relative to the deviation of the ratios to the Vienna Standard Mean Ocean Water (V-SMOW).

$$\delta^{18}O = \left[ \frac{R_{sample}}{R_{standard}} - 1 \right] \times 1000‰ \tag{1}$$

$$\delta^2H = \left[ \frac{R_{sample}}{R_{standard}} - 1 \right] \times 1000‰ \tag{2}$$

where $R_{sample}$ is the ratio of $^{18}O/^{16}O$ ($^2H/^1H$) in the water sample, and $R_{standard}$ is the ratio of $^{18}O/^{16}O$ ($^2H/^1H$) in V-SMOW [52]. The measurement accuracy for $\delta^{18}O$ and $\delta^2H$ is 0.3‰ and 1‰, respectively.

### 2.5. Linear Mixing Model and Bayesian Mixing Model

The Linear mixing model is solved by linear equations. Assuming that the precipitation is a mixture of advected, evaporated and transpired moisture, the recycling fractions can be calculated by [38]:

$$\delta^{18}O_S = F_{adv}\delta^{18}O_{adv} + F_{evap}\delta^{18}O_{evap} + F_{tr}\delta^{18}O_{tr} \tag{3}$$

$$d_s = F_{adv}d_{adv} + F_{evap}d_{evap} + F_{tr}d_{tr} \tag{4}$$

$$1 = F_{adv} + F_{evap} + F_{tr} \tag{5}$$

where $F_{adv}$, $F_{evap}$ and $F_{tr}$ are the fractions of advected moisture, evaporated moisture and transpired moisture, respectively; $\delta^{18}O_S$, $\delta^{18}O_{adv}$, $\delta^{18}O_{evap}$ and $\delta^{18}O_{tr}$ are the compositions of $\delta^{18}O$ for local subcloud precipitation, advected, evaporated and transpired moisture, respectively; and $d_s$, $d_{adv}$, $d_{evap}$ and $d_{tr}$ are the D-excess of local subcloud precipitation, advected, evaporated and transpired moisture, respectively.

The Bayesian mixing model is performed in MixSIAR GUI. The MixSIAR GUI is a graphical user interface (GUI) that helps user create and run Bayesian mixing models to analyze biotracer data following the MixSIAR model framework. Both the GUI and script versions are written in the open-source languages R [53], then MixSIAR writes a custom JAGS (Just Another Gibbs Sampler, Plummer) [54] model file; runs the model in JAGS; and produces diagnostics, posterior plots and summary statistics [55]. It should be noted that MixSIAR is based on MCMC (Markov chain Monte Carlo) chains, so before using the output, one should be able to confirm that the chain is converged by one of the two default methods for diagnosing the convergence of the model, namely Gelman–Rubin and Geweke diagnoses.

More information about MixSIAR GUI was introduced by Stock and Semmens [46], and for most of the mathematical formulation underlying model and primary citation for Bayesian mixing models, refer to [41,45].

### 2.6. Calculations

2.6.1. Isotope Composition in Precipitation Vapor ($\delta_{pv}$)

The isotope composition in precipitation vapor ($\delta_{pv}$) was calculated using the software Hydrocalculator [56] using measured isotopes in precipitation ($\delta_p$) corrected by local evaporation line (LEL) [57]:

$$\delta_{pv} = \frac{\delta_p - x\varepsilon^+}{1 + 10^{-3} \times x\varepsilon^+} \tag{6}$$

where $\delta_p$ is isotopic composition in precipitation, $x$ is an adjusting parameter [58], the calculations were repeated using $x$ between 0.6 and 1.0 with step width of 0.0001; the final value of $x$ was when the difference between the calculated $Slope_{LEL}$ and the actual LEL slope was the lowest or $x$ reached the boundary values (0.6 or 1.0). $\varepsilon^+$ is the temperature dependent equilibrium fractionation factor defined as:

$$\varepsilon^+ = 1000 \times \left(\alpha^+ - 1\right) \tag{7}$$

and $\alpha^+$ is given by [59]:

$$10^3 ln^2\alpha^+ = \frac{1158.8T^3}{10^9} - \frac{1620.1T^2}{10^6} + \frac{794.84T}{10^3} - 161.04 + \frac{2.9992 \times 10^9}{T^3} \tag{8}$$

$$10^3 ln^{18}\alpha^+ = -7.685 + \frac{6.7123 \times 10^3}{T} - \frac{1.6664 \times 10^6}{T^2} + \frac{0.35041 \times 10^9}{T^3} \tag{9}$$

where $T$ is given in Kelvin degrees.

The measured *LEL* slope was derived from pan evaporation experiments. $Slope_{LEL}$ is calculated by [56]:

$$Slope_{LEL} = \frac{\left[\frac{h \times \left(10^{-3} \times \delta_{pv}^2 - 10^{-3} \times \delta_p^2\right) + \left(1 + 10^{-3} \times \delta_p^2\right) \times 10^{-3} \times \varepsilon}{h - 10^{-3} \times \varepsilon}\right]_H}{\left[\frac{h \times \left(10^{-3} \times \delta_{pv}^{18} - 10^{-3} \times \delta_p^{18}\right) + \left(1 + 10^{-3} \times \delta_p^{18}\right) \times 10^{-3} \times \varepsilon}{h - 10^{-3} \times \varepsilon}\right]_O} \tag{10}$$

where $h$ is relative humidity, $\delta_{pv}$ and $\delta_p$ are isotope composition in precipitation vapor and precipitation, and $\varepsilon$ is the total fractionation factor defined as [60]:

$$\varepsilon = \frac{\varepsilon^+}{\alpha^+} + \varepsilon_k \tag{11}$$

where $\varepsilon^+$ and $\alpha^+$ was calculated by Equations (7)–(9), and $\varepsilon_k$ is the kinetic fractionation factor [61], which was calculated by:

$$\varepsilon_k = (1 - h)C_k \tag{12}$$

where $h$ is relative humidity, the value of the kinetic fractionation constant $C_k$ is 12.5‰ for $\delta^2 H$ and 14.2‰ for $\delta^{18}O$, respectively [62,63].

### 2.6.2. Isotope Composition in Advected Vapor ($\delta_{adv}$)

The isotope composition in advected vapor ($\delta_{adv}$) was calculated by the Rayleigh distillation equation:

$$\delta_{adv} = \delta_{pv\text{-}adv} + \left(\alpha^+ - 1\right) lnF \tag{13}$$

where $\delta_{pv\text{-}adv}$ is the isotope composition in precipitation vapor at the upwind station and can be calculated using Equation (6), $\alpha^+$ is calculated using Equations (8) and (9) and $F$ is a ratio between final and initial vapor. It is difficult to acquire a ratio between final and initial vapor; we used the surface vapor pressure for each station to calculate the value of $F$ [39]. The monthly mean vapor pressure data was provided by the National Meteorological Information Center of China (http://data.cma.cn/) (accessed on 11 March 2019).

### 2.6.3. Isotope Composition in Transpired Vapor ($\delta_{tr}$)

Generally, the isotope compositions in plants are unfractionated during water uptake by roots until it arrives at leaf [64–66]. Hence, the isotope composition in transpired vapor ($\delta_{tr}$) can be determined based on the plant xylem water isotope composition. However, it is difficult to determine which plants represent the average transpiration and how they distribute contribution ratios. Therefore, weighted average precipitation is usually used instead of plant xylem water [39]:

$$\delta_{tr} \approx \delta_p \tag{14}$$

### 2.6.4. Isotope Composition in Surface Evaporated Vapor ($\delta_{ev}$)

The Craig–Gordon model [67–69] was used to estimate the isotope composition in surface evaporated vapor ($\delta_{ev}$):

$$\delta_{ev} = \frac{\frac{\delta_s}{\alpha^+} - h\delta_{adv} - \varepsilon}{1 - h + \varepsilon_k} \tag{15}$$

where $\delta_s$ is the isotope composition of liquid water at the evaporating front; the soil water content (SWC)-weighted isotope composition of soil water was used in this study. $\delta_{adv}$ is the isotope composition of advected vapor, $h$ is relative humidity and $\alpha^+$ is calculated by Equations (8) and (9). $\varepsilon$ is the total fractionation factors calculated by Equation (11), and

$\varepsilon_k$ is the kinetic fractionation factor. The kinetic fractionation factor ($\varepsilon_k$) for soil surface evaporation can be calculated by [69]:

$$\varepsilon_k = (1 - h)\theta n C_D \tag{16}$$

where $h$ is relative humidity; the weighting value $\theta$ for small water body (evaporation flux does not influence ambient humidity) is 1, and for larger water body (like the Great Lakes and the Mediterranean Sea) is 0.5; the value $n$ for a stable layer (like soil and leaf cover) is 1, and for a large open water body is 0.5; the values of $\theta$ and $n$ are both set to 1 in this study; the values of $^2C_D$ and $^{18}C_D$ are 25.1‰ and 28.5‰, respectively [63,70].

## 3. Results

### 3.1. Percent Contributions of Different Water Vapor to Local Precipitation

Table 4 and Figure 2 show the calculated percent contributions of different water vapor to local precipitation and the process of precipitation formation (Supplementary Material). It can be seen that the local precipitation during the growing season in the Qilian Mountains was mainly supplied by advected water vapor. It should be noted that moisture added to the atmosphere by evaporation and transpiration may travel hundreds or thousands of miles before being re-precipitated [9], so it is not always from precipitation in the same place. Even if it precipitated in the same place, the moisture returned to the atmosphere by continental evapotranspiration may be absorbed by dry continental air mass that, in general, do not produce rainfall in arid regions [8]. However, despite that, the contribution of recycled moisture to local precipitation cannot be ignored.

**Table 4.** Average percent contributions of the three water vapor-to-precipitation sources using MixSIAR.

| Sites | Month | $f_{adv}$/% | $f_{ev}$/% | $f_{tr}$/% |
|-------|-------|-------------|------------|------------|
| Sunan | May | 53 | 19 | 28 |
| | June | 57 | 19 | 24 |
| | July | 57 | 24 | 19 |
| | August | 70 | 15 | 15 |
| | September | 57 | 19 | 24 |
| Gulang | May | 63 | 17 | 20 |
| | June | 68 | 15 | 17 |
| | July | 51 | 23 | 26 |
| | August | 72 | 15 | 13 |
| | September | 77 | 11 | 12 |
| Yeniugou | May | 66 | 15 | 19 |
| | June | 73 | 13 | 14 |
| | July | 43 | 26 | 31 |
| | August | 68 | 16 | 16 |
| | September | 65 | 17 | 18 |
| Wushaoling | May | 53 | 22 | 25 |
| | June | 69 | 15 | 16 |
| | July | 46 | 24 | 30 |
| | August | 74 | 13 | 13 |
| | September | 76 | 12 | 12 |
| Tianzhu | May | 68 | 15 | 17 |

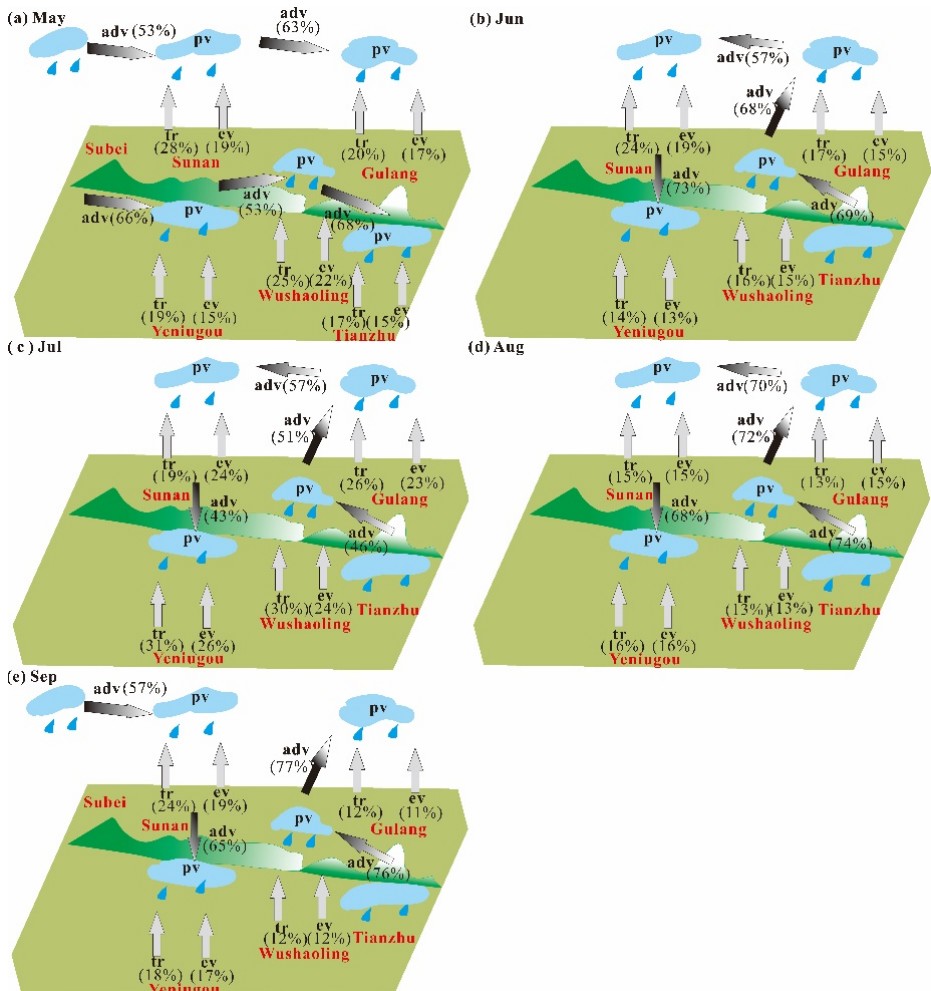

**Figure 2.** Simulation of the precipitation formation at the Qilian Mountains in the plant growing season of 2017, (**a–e**) represents May to September, respectively.

The contribution of recycled moisture to precipitation at the Qilian Mountains was average 37%, and the contribution of transpirated vapor was generally bigger than evaporated vapor; this showed the importance of vegetation transpiration in such an alpine area. On average, the contribution of recycled moisture to precipitation at west sites were bigger than the east sites, and the north sites were bigger than the south sites.

The result of 37% was bigger than that calculated by Li et al. [38] for the Shiyang river basin (23%). This was associated with the spatial heterogeneity of the study region and the calculation procedure, especially for isotope ratio in advected vapor. The isotope composition in advected vapor in this paper was based on the measured precipitation isotopes at the upwind stations using the Rayleigh distillation equation. However, upwind stations were not used by Li et al. [38], and the D-excess and $\delta^{18}O$ in the growing season at the potential upwind were estimated using the observations during the cold winter period, as well as the linear regression between isotopic composition and temperature. Logically, the winter and summer vapor do not always share the same trajectory, and the temperature effect varies in different seasons.

The results calculated by MixSIAR were presented by mean (standard deviation) and the quartiles (2.5%, 5%, 25%, 50%, 75%, 95%, 97.5%), not the whole original feasible solutions. Here, the upper quartile, the lower quartile and the median values were plotted as boxes (Figure 3). In the plant growing season, regardless of the month, the dominant position of advected vapor was obvious. The contribution of transpirated vapor was the largest in July at Gulang, Yeniugou and Wushaoling, except at Sunan, which was in May. It

should be noted that there still exists uncertainty to the estimate results of the three types of water vapor to precipitation, because the results of Bayesian estimation largely depend on the prior probability, and it is not complete acceptance or rejection of the hypothesis; hence, there exists uncertainty to average 37% (the upper quartile and the lower quartile was 30% and 43%, respectively) for the contribution of recycled moisture to precipitation at the Qilian Mountains.

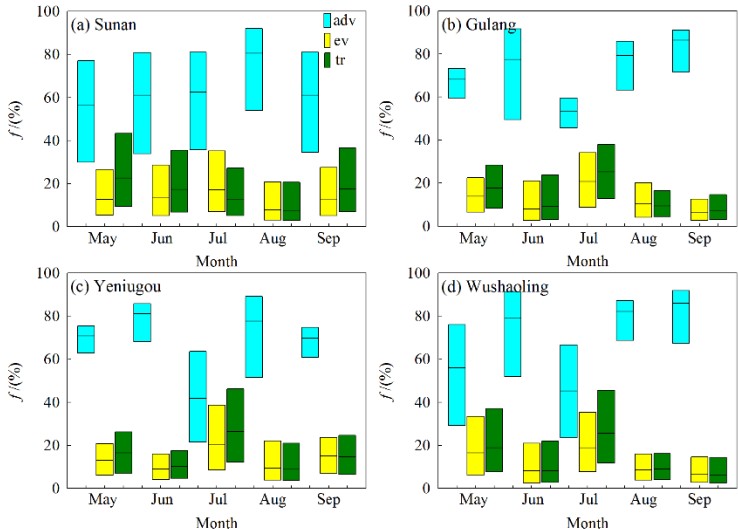

**Figure 3.** Box plot of monthly percent contributions of the three water vapor-to-precipitation methods. In the box plot, the rectangles span the first quartile to the third quartile, and the segment inside the rectangle shows the median value, (**a**–**d**) represents Sunan, Gulang, Yeniugou and Wushaoling, respectively.

### 3.2. Isotopic Characteristics of Precipitation and Each Water Vapor

The monthly mean values of precipitation vapor and the three sources of vapor are presented in Figure 4. It can be seen that the isotope composition of precipitation was more enriched than precipitation vapor; this was because the subcloud evaporation increased the value of $\delta^{18}O$ and $\delta^{2}H$ in precipitation. The isotope composition of precipitation vapor was close to that of advected vapor, suggesting the main vapor source was external. Since subcloud evaporation increased the isotope value in precipitation, and recycled moisture reduced the isotope value in precipitation [38], the final isotope value of precipitation falling to the ground basically ranged between the value of precipitation vapor (close to advected vapor) and the three component vapors.

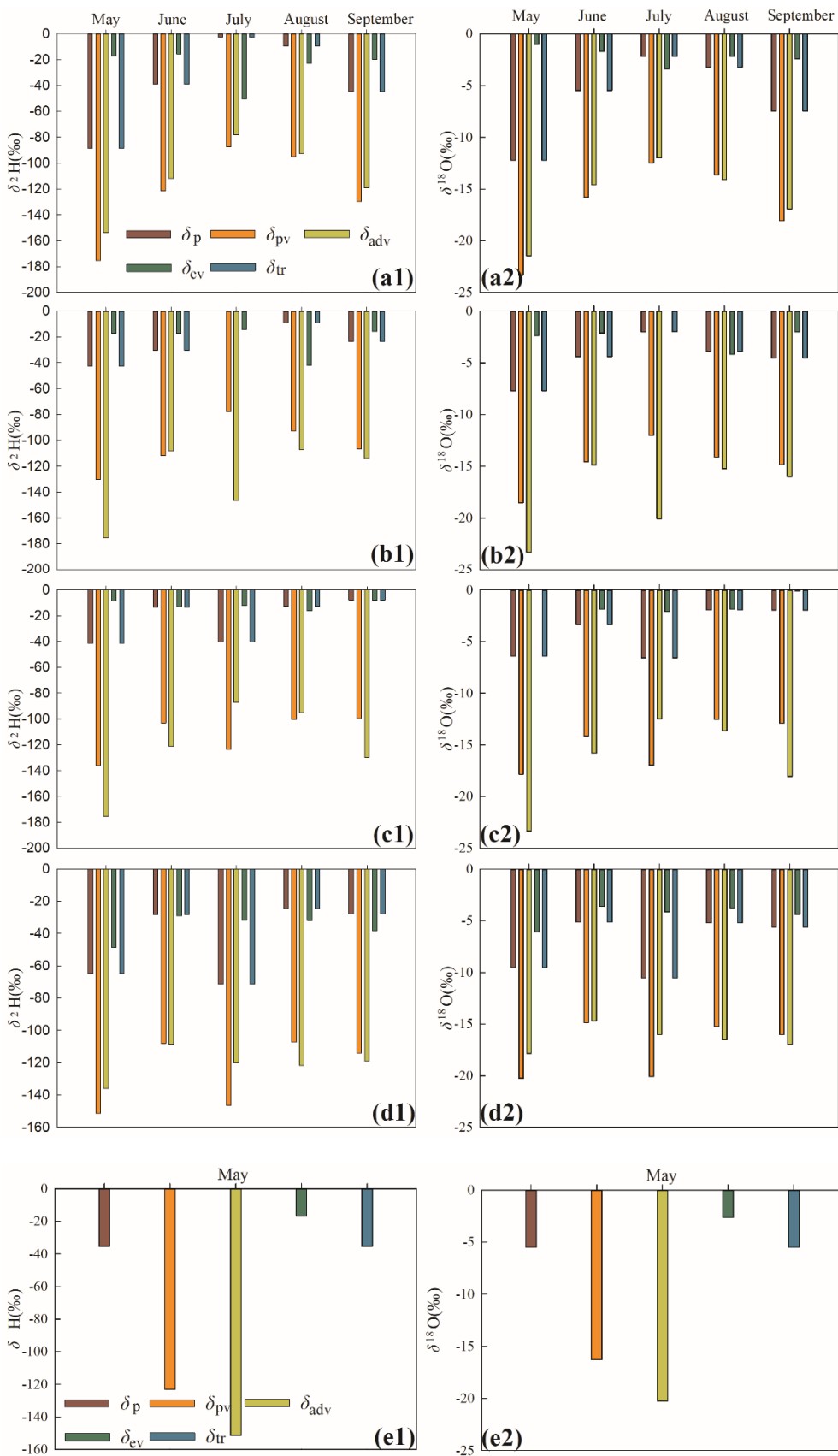

**Figure 4.** Monthly isotope composition of $\delta_p$-precipitation, $\delta_{pv}$-precipitation vapor, $\delta_{adv}$-advected vapor, $\delta_{ev}$-evaporated vapor and $\delta_{tr}$-transpirated vapor (**a1,a2**-Sunan; **b1,b2**-Gulang; **c1,c2**-Yeniugou; **d1,d2**-Wushaoling; **e1,e2**-Tianzhu).

## 4. Discussion

Although it is not easy to accurately estimate or measure isotope ratios in each end member to local precipitation, a significant number of case studies have focused on this topic in recent decades. For a small ecosystem scale, it is effective to directly measure or easily estimate the isotopic variation in water vapor using a laser-based online analyzer [71–73]. However, due to long distance, this procedure seems not helpful on a larger spatial scale, such as a trajectory for several hundred kilometers.

The temporal resolution is an important issue on the calculation of recycling ratio using an isotopic method. In many previous studies (e.g., [5,39]), the recycling ratios were expressed as a mean value of the plant growing season, and some important details on a monthly scale were ignored. During the growing season for several months, the percent contribution for each flux is not a constant. In this study, each month during the growing season was calculated respectively, which may be useful to understand the seasonality of hydrological processes in such an alpine setting.

Regarding the widely applied three-end-member linear mixing model, the calculation does not always make sense, and the equations may have no solution in many cases. In this study, when analyzing the percent contributions of transpired, evaporated and advected vapor to precipitation using the three-end-member linear mixing model, there were many percent contributions more than 100% or smaller than 0% (Table 5), and the average contribution of recycled moisture is negative (−5%).

**Table 5.** Percent contributions of the advected vapor ($f_{adv}$), evaporated vapor ($f_{ev}$) and transpired vapor ($f_{tr}$) to precipitation using linear mixing model in the Qilian Mountains.

| Sites | Months | Three-End-Member | | | Two-End-Member | | |
|---|---|---|---|---|---|---|---|
| | | $f_{adv}$/% | $f_{ev}$/% | $f_{tr}$/% | $f_{adv}$/% | $f_{ev}$/% | $f_{tr}$/% |
| Sunan | 5 | 240 | 99 | −239 | 109 | −9 | / |
| | 6 | 112 | −2 | −10 | 109 | −9 | / |
| | 7 | 103 | 14 | −17 | 105 | 5 | / |
| | 8 | 98 | 28 | −26 | 96 | 4 | / |
| | 9 | 119 | 13 | −32 | 108 | −8 | / |
| Gulang | 5 | 62 | −21 | 59 | 69 | / | 31 |
| | 6 | 128 | 142 | −170 | 98 | 2 | / |
| | 7 | 55 | −3 | 48 | 55 | / | 45 |
| | 8 | 90 | −15 | 25 | 90 | / | 10 |
| | 9 | 93 | 16 | −9 | 92 | 8 | / |
| Yeniugou | 5 | 76 | 23 | 1 | 76 | 23 | 1 |
| | 6 | 83 | −31 | 48 | 87 | / | 13 |
| | 7 | 183 | 9 | −92 | 143 | −43 | / |
| | 8 | 93 | 334 | −327 | 91 | 9 | / |
| | 9 | 75 | 63 | −38 | 71 | 29 | / |
| Wushaoling | 5 | 113 | −39 | 26 | 129 | / | −29 |
| | 6 | 100 | −13 | 13 | 102 | / | −2 |
| | 7 | 109 | −56 | 47 | 174 | / | 74 |
| | 8 | 86 | −18 | 32 | 89 | / | 11 |
| | 9 | 93 | 13 | −6 | 92 | 8 | / |
| Tianzhu | 5 | 87 | 75 | −62 | 77 | 23 | / |

We considered an alternative method for this condition, i.e., for the flux with a proportion greater than 100%, the vapor with the smallest contribution (i.e., the most negative) was considered to be negligible, and then the contribution was calculated again using the two-end-member linear mixing model without the negligible flux. Although this time a reasonable result was obtained for 9%, there were still contributions greater than 100% or less than 0% (Table 5), so the linear mixing model is not ideal or we need more parameter adjustment, at least. This is why the Bayesian estimation should be recommended on the

recycling ratio calculation. At least it works better than the usually applied simple linear mixing model.

Regarding the isotope composition in advection, the practical methods are usually quite different in previous publications. For some regions, if the dominant upwind direction is generally constant, the marginal stations at the upwind can be selected to calculate the advection isotopes [39,50]. When the sampling stations are not enough, the isotopes in growing season were estimated using the measurement of precipitation in nongrowing season without much evapotranspiration, in many cases [6,38]. However, in a region with multiple moisture paths, such as the study region jointly controlled by midaltitude westerly and monsoon, the default upwind station seems not suitable in the modeling. In this study, a varying upwind station setting was applied, which is generally based on the meteorological diagnostic using global reanalysis. In this assumption with different moisture trajectories, the moisture transport is logically more suitable in the Qilian Mountains at the marginal area of Asian monsoon. If only the westerly or monsoon moisture is considered as the dominant upwind, it is not consistent with the meteorological and climate background in this region [74].

For the isotope composition of transpiration, the seasonal and diurnal variation of water isotope in plant xylem was usually large, and different plant species resulted in different xylem water isotopes as evidenced in previous studies [75–77]. It is not possible to collect all plant species, and the stable isotope ratios in transpired vapor are not always similar to that in xylem water, especially in non-steady-state or some arid condition [78,79]. Even if we can sample them, which well represents the local ecosystem, weighting for each species is another issue. In central Asia, Wang et al. [39] indicated that the isotope values of precipitation and xylem water have good consistency and recommended that the weighted average precipitation isotope values can be used instead of the xylem water. For evaporated vapor, the SWC-weighted soil water isotopes were used, and this is better than the assumption that the stable isotopes in surface water equals to those in long-term precipitation in previous studies.

Just as the findings of this paper, some studies have shown that the recycled moisture in arid and semiarid regions may be the most important source of rainfall [7]. However, it should be noted that only the vapor contributions during the growing season were investigated in this paper, and the contributions of evapotranspiration and advection may be converted during different seasons [1]. If the research is carried out in other months of the year, i.e., nongrowing season, the two-end-member mixing model is appropriate, because there are few contributions of plant transpiration to precipitation when it is in a nongrowing season period. In addition, percent contribution of transpired vapor was bigger than evaporated vapor in the growing season, but the results may be different if the sample number increased by the collection of surface water, such as lake water, reservoir water or irrigation water in a nongrowing season. Because the content of advected vapor in other months is relatively less than in the growing season (precipitation event mainly occurred in this period), hence the evaporated vapor will active to make contribution to precipitation, it is likely that the percent contribution of evaporated vapor will increase significantly.

## 5. Conclusions

The contribution of recycled moisture to precipitation is an important part of water cycle. The Bayesian model was first used to calculate the contribution of different water vapor to local precipitation. The commonly used software IsoError may produce results greater than 100% or less than 0%, which needs further parameter adjustment, and may underestimate the contribution rate, while the Bayesian model is based on the MCMC method, and, although it is not perfect, either, it describes the phenomena better than a simpler linear mixing model. These two methods have different parameter estimation; more accurate quantification of the contribution of recycled moisture is particularly important

for clarifying water vapor contribution of vegetation transpiration and surface evaporation, especially in arid, semiarid and alpine regions.

The results showed that the average contribution of recycled moisture (including surface evaporated vapor and plant transpirated vapor) to precipitation was 37% (the upper quartile and the lower quartile were 30% and 43%, respectively) during the plant growing season in 2017 at the Qilian Mountains, which is an important part of precipitation formation in alpine region.

**Supplementary Materials:** The following are available online at https://www.mdpi.com/article/10.3390/atmos12060731/s1, Figure S1: Monthly mean precipitable water (PW) and vapor flux (VF) in the Qilian Mountains in May, June, July, August and September in 2017, Table S1: Determination of upwind stations in the Qilian Mountains from May to September in 2017.

**Author Contributions:** Conceptualization, X.Q. and S.W.; methodology, X.Q.; software, X.Q. and S.W.; validation, M.Z.; investigation, X.Q., X.Y., H.M. and C.C.; data curation, M.Z.; writing, X.Q. and Z.D.; visualization, X.Q.; supervision, M.Z. and S.W.; project administration, M.Z.; funding acquisition, M.Z. All authors have read and agreed to the published version of the manuscript.

**Funding:** This research was funded by National Natural Science Foundation of China, grant number 41461003; Scientific Research Program of Higher Education Institutions of Gansu Province, grant number 2018C-02; Foundation of Key Laboratory for Ecology and Environment of River Wetlands in Shaanxi Province, grant number SXSD201703.

**Institutional Review Board Statement:** Not applicable.

**Informed Consent Statement:** Not applicable.

**Data Availability Statement:** The data applied in this study is available from the authors.

**Acknowledgments:** The authors greatly thank all the meteorological stations for their warm help in field sampling. We also thank the colleagues in the Northwest Normal University for in laboratory analyzing and field work, especially Su'e Zhou and Yaning Zhang.

**Conflicts of Interest:** The authors declare no conflict of interest.

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
