# Peer review of "Contribution of Recycled Moisture to Precipitation in Northeastern Tibetan Plateau: A Case Study Based on Bayesian Estimation"

_atmosphere, doi:10.3390/atmos12060731_

Round 1

Reviewer 1 Report

The article is well structured with the objectives of the work well expressed and contextualized. The work is based on a well-explained and defined method. The results are compliant and discussed through the available bibliography. In general, what I would ask is to dwell more on the fact that the data used refers to a single investigative season and therefore does not take into account seasonal and annual climatic variations. For this reason they cannot represent a general contribution calculation applicable to different time scales. Furthermore, I would ask the authors to explain if the season in question is representative of the general (or seasonal) conditions of the area and if it is not some sort of anomalous case in terms of environmental, climatic, vegetative conditions, etc.

Here are some specific references:

Line 38: coefficient of measure units is wrong: “500 km2 scales is 9.6% and for 1000 km2”. Check it in all text.

Figure 1: Is it possible to make all the writings more legible? Both the toponyms and the numbers of the scale bar are difficult to read. Perhaps introducing buffers around the writings could have more visual clarity. In the box at the top right it is not possible to understand the dimensionality of the map. Is it possible to add a scale or geographic coordinates? The desert areas are not very visible and blend in with the background. Is it possible to thematize them in a different way so that they are clearer?

2.2 Data: Why was it chosen to use only the year 2017? Unique available? the choice is clear (due to sampling) but only after reading the next chapter. I would recommend reversing at least chapters 2.2 and 2.3 so that the choice of data is clearer to the reader.

Reviewer 2 Report

Please, find my comments in the attached file.

Reviewer 3 Report

Because Qillian Mountains are the region where the significant climatic and environmental changes have been observed, the investigation of local moisture recycling is crucial to manage the water resources. So this study is valuable. It has tried to investigate the water source for the precipitation using the Bayesian mixing model for the first time. Please see below for some minor comments.

-Even though the location of region such as Subei, Sunan, Gulang in Table 4 is marked in Figure 1, itis better to mark in Figure 2 again to follow the wind direction noted in Table 4.

-Line 287-290: Because the author mention that the west and north sites show bigger transpiration than the other sites, It would better to show the differences in the vegetation amount over the sites (north vs. south and east vs. west).

-Captions for Figure 7 and Figure 8 are same. Need to be corrected.

Round 2

Reviewer 2 Report

The authors have modified the paper following the suggestions proposed in my first review so I recommend the publication of the article.

Author Response

There was no response to reviewer 2' s comments